# Sensitivity of Stability: Theoretical & Empirical Analysis of Replicability for Adaptive Data Selection in Transfer Learning

## Abstract

The widespread adoption of transfer learning has revolutionized machine learning by enabling efficient adaptation of pre-trained models to new domains. However, the reliability of these adaptations remains poorly understood, particularly when using adaptive data selection strategies that dynamically prioritize training examples. We present a comprehensive theoretical and empirical analysis of replicability in transfer learning, introducing a mathematical framework that quantifies the fundamental trade-off between adaptation effectiveness and result consistency. Our key contribution is the formalization of *selection sensitivity* ($\Delta_Q$), a measure that captures how adaptive selection strategies respond to perturbations in training data. We prove that replicability failure probability: the likelihood that two independent training runs produce models differing in performance by more than a threshold, increases quadratically with selection sensitivity while decreasing exponentially with sample size. Through extensive experiments on the MultiNLI corpus using six adaptive selection strategies—ranging from uniform sampling to gradient-based selection—we demonstrate that this theoretical relationship holds precisely in practice. Our results reveal that highly adaptive strategies like gradient-based and curriculum learning achieve superior task performance but suffer from high replicability failure rates, while less adaptive approaches maintain failure rates below 7%. Crucially, we show that source domain pretraining provides a powerful mitigation mechanism, reducing failure rates by up to 30% while preserving performance gains. These findings establish principled guidelines for practitioners to navigate the performance-replicability trade-off and highlight the need for replicability-aware design in modern transfer learning systems.

## 1 Introduction

The ability to reliably reproduce experimental results lies at the heart of scientific inquiry. In machine learning, this fundamental principle faces unprecedented challenges as models grow in complexity and training procedures become increasingly sophisticated Henderson et al. (2019); Impagliazzo et al. (2023); Beam et al. (2020); Kapoor & Narayanan (2023). The recent surge in transfer learning applications—where pre-trained models are adapted to new domains—has amplified these concerns, as practitioners routinely employ adaptive data selection strategies that dynamically shape the training process based on model state and data characteristics Jiang et al. (2024). Recent surveys indicate that over 70% of ML researchers struggle to replicate published results, highlighting the severity of this reproducibility crisis Desai et al. (2025); Hutson (2018).

Transfer learning has emerged as the dominant paradigm for deploying machine learning systems across diverse applications, from natural language processing to computer vision Zhu et al. (2020); Chen et al. (2020). By leveraging knowledge from large-scale pre-training, these approaches achieve remarkable performance even with limited domain-specific data. However, this efficiency comes with hidden costs: the adaptive mechanisms that make transfer learning effective also introduce substantial variability in outcomes. Recent studies have documented cases where identical experimental setups yield significantly different results

across runs, raising critical questions about the reliability of reported findings Bouthillier et al. (2019); Kou (2024).

At the core of this challenge lies a fundamental tension between adaptation and stability. Adaptive data selection strategies—including importance weighting, confidence-based sampling, curriculum learning, and gradient-based selection—improve learning efficiency by focusing computational resources on the most informative examples Ren et al. (2019). Recent advances in adaptive learning platforms demonstrate the practical benefits of these approaches Strielkowski et al. (2025). Yet this very adaptivity creates complex dependencies between the selection mechanism and the evolving model state, potentially amplifying small variations in initialization or data ordering into substantial differences in final performance. Understanding and quantifying this trade-off has become essential as machine learning systems are deployed in high-stakes applications where consistency and reliability are paramount.

The theoretical foundations for understanding replicability in machine learning have advanced significantly through connections to algorithmic stability, differential privacy, and statistical learning theory Bun et al. (2023); Chase et al. (2023). These frameworks establish that replicability—defined as the property that an algorithm produces consistent outputs when applied to independent samples from the same distribution—is intimately connected to how algorithms respond to perturbations in their inputs. Recent work has begun exploring replicability in specific contexts, such as online learning Ahmadi et al. (2024) and reinforcement learning Eaton et al. (2023). However, existing theory has not adequately addressed the dynamic, adaptive nature of modern transfer learning pipelines.

Our work bridges this gap by developing a comprehensive framework for analyzing replicability in transfer learning with adaptive data selection. We introduce the concept of *selection sensitivity* ($\Delta_Q$), which quantifies how much a selection strategy's distribution changes in response to small perturbations in the training data. This measure provides a principled way to compare different adaptive strategies and predict their impact on replicability. Our theoretical analysis reveals that the probability of replicability failure scales quadratically with selection sensitivity, providing tight bounds that enable practitioners to make informed decisions about strategy selection based on their tolerance for variability. Here, replicability failure refers to the event that two models trained independently on different samples from the same distribution produce performance differences exceeding a small threshold—a notion distinct from the algorithmic replicability , which typically requires syntactic equivalence of outputs rather than statistical consistency of performance.

Through extensive empirical evaluation on six distinct selection strategies—ranging from simple uniform sampling to sophisticated approaches—we validate our theoretical predictions and uncover practical insights for improving replicability without sacrificing performance. Our experiments on the MultiNLI corpus demonstrate that while highly adaptive strategies can improve accuracy, they also increase replicability failure rates from 2.2% (uniform baseline) to over 80% for gradient based methods in the two-stage fine-tuning setting. Most significantly, we show that incorporating source domain pretraining before adaptive fine-tuning provides a robust mitigation strategy, dramatically improving replicability while maintaining the performance benefits of adaptive selection.

This paper addresses the following research questions:

> **RQ1:** *How can we formally characterize and quantify the impact of adaptive data selection strategies on the replicability of transfer learning outcomes?*

> **RQ2:** *What is the precise mathematical relationship between selection sensitivity, sample size, and the probability of replicability failure in transfer learning?*

> **RQ3:** *How do different families of adaptive selection strategies—from simple importance weighting to complex gradient-based methods—compare in terms of their selection sensitivity and resulting impact on replicability?*

> **RQ4:** *What practical mitigation strategies can reduce replicability failures while preserving the performance benefits of adaptive selection in real-world transfer learning scenarios?*

Our contributions advance both theoretical understanding and practical application of replicable machine learning:

1. We develop a rigorous mathematical framework that formalizes replicability in transfer learning with adaptive data selection, introducing selection sensitivity ($\Delta_Q$) as a key measure of algorithmic stability.
2. We prove tight theoretical bounds showing that replicability failure probability scales as $\rho \leq 4\exp(-\epsilon^2 n/2c^2\Delta_Q^2)$, revealing the quadratic impact of selection sensitivity and exponential improvement with sample size.
3. We provide comprehensive empirical validation across six selection strategies on the MultiNLI corpus, demonstrating that our theoretical predictions accurately capture real-world behavior across a wide range of sensitivity values .
4. We identify source domain pretraining as a powerful mitigation strategy that reduces replicability failure rates by up to 30% while maintaining performance gains, offering a practical solution for reliability-critical applications.
5. We release a complete experimental framework [1] that enables researchers to evaluate the replicability of their own transfer learning approaches and explore new selection strategies within our theoretical framework.

The remainder of this paper is organized as follows. Section 2 establishes the theoretical foundations, formalizing transfer learning, adaptive selection, and replicability. Section 3 presents our main theoretical results, including the stability lemma and replicability bounds. Section 4 describes our experimental setup, including the six selection strategies and evaluation methodology. Section 5 presents comprehensive empirical results validating our theory and exploring practical implications. Section 6 discusses broader implications and connections to related work. Finally, Section 7 addresses limitations and future directions.

## 2 Theoretical Foundations

In this section, we establish the theoretical framework for analyzing replicability in transfer learning with adaptive data selection strategies. We present formal definitions of transfer learning, adaptive data selection, and replicability, which will form the basis for our theoretical analysis and empirical evaluation.

### 2.1 Transfer Learning Framework

We begin with a formal definition of the transfer learning setup following the notation established in Pan & Yang (2010).

**Definition 1** (Domain)**.** *A domain $\mathcal{D}$ consists of a feature space $\mathcal{X}$ and a marginal probability distribution $P(X)$, where $X = \{x_1, \ldots, x_n\} \subset \mathcal{X}$. We denote a domain as $\mathcal{D} = \{\mathcal{X}, P(X)\}$.*

**Definition 2** (Task)**.** *Given a domain $\mathcal{D}$, a task $\mathcal{T}$ consists of a label space $\mathcal{Y}$ and a conditional probability distribution $P(Y|X)$ that is typically learned from training data consisting of pairs $\{x_i, y_i\}$, where $x_i \in \mathcal{X}$ and $y_i \in \mathcal{Y}$. We denote a task as $\mathcal{T} = \{\mathcal{Y}, P(Y|X)\}$.*

**Definition 3** (Transfer Learning)**.** *Given a source domain $\mathcal{D}_S$ with task $\mathcal{T}_S$ and a target domain $\mathcal{D}_T$ with task $\mathcal{T}_T$, transfer learning aims to improve the learning of the target predictive function $f_T(\cdot)$ in $\mathcal{T}_T$ using the knowledge in $\mathcal{D}_S$ and $\mathcal{T}_S$, where $\mathcal{D}_S \neq \mathcal{D}_T$ or $\mathcal{T}_S \neq \mathcal{T}_T$.*

In the context of our work, we focus on a common transfer learning scenario where the feature spaces are the same ($\mathcal{X}_S = \mathcal{X}_T$), the label spaces are the same ($\mathcal{Y}_S = \mathcal{Y}_T$), but the marginal distributions differ ($P_S(X) \neq P_T(X)$). This setting is known as domain adaptation Blitzer et al. (2006).

We formalize the learning problem as follows. Let $\mathcal{H}$ be a hypothesis class (set of possible models) and $L : \mathcal{H} \times \mathcal{X} \times \mathcal{Y} \to [0,1]$ be a loss function. We assume access to a source sample $S = \{(x_i^s, y_i^s)\}_{i=1}^{n_S} \sim \mathcal{D}_S^{n_S}$ and

---

[1] `code-to-be-released-post-review`

a target sample $T = \{(x_i^t, y_i^t)\}_{i=1}^{n_T} \sim \mathcal{D}_T^{n_T}$. The goal is to learn a model $h_T \in \mathcal{H}$ that minimizes the expected risk on the target domain:

$$R_T(h) = \mathbb{E}_{(x,y)\sim\mathcal{D}_T}[L(h, x, y)] \tag{1}$$

In practice, we typically have a pre-trained model $h_0$ from the source domain, and we fine-tune it on the target data to obtain $h_T$. The effectiveness of this process depends significantly on the sample selection strategy used during fine-tuning.

## 2.2 Adaptive Data Selection

Traditional fine-tuning methods use all available target data with uniform importance. In contrast, adaptive data selection strategies dynamically adjust the importance of different training examples based on various criteria. We formalize this as follows:

**Definition 4** (Selection Strategy). *A selection strategy $Q$ is a function that assigns a probability distribution over the training examples. For a training set $T = \{(x_i, y_i)\}_{i=1}^n$, the strategy $Q$ assigns weights $Q(x_i, y_i)$ to each example such that $\sum_{i=1}^n Q(x_i, y_i) = 1$ and $Q(x_i, y_i) \geq 0$.*

The selection strategy $Q$ can be a function of the current model state, the training history, or properties of the data itself. This leads to an effective empirical risk minimization objective:

$$R_T(h; Q) = \sum_{i=1}^{n_T} Q(x_i, y_i) L(h, x_i, y_i) \tag{2}$$

We now present mathematical formulations of six common selection strategies[2]:

### 2.2.1 Uniform Strategy

The uniform strategy, which serves as our baseline, assigns equal weight to all examples. This is equivalent to standard empirical risk minimization, where all examples contribute equally to the objective.

$$Q_{uniform}(x_i, y_i) = \frac{1}{n}, \forall i \in \{1, \ldots, n\} \tag{3}$$

### 2.2.2 Importance Weighting Strategy

Importance weighting addresses domain shift by assigning weights based on the similarity between source and target distributions Cortes et al. (2010); Shimodaira (2000). In our context, we weight examples based on domain features (e.g., genre or document type):

$$Q_{IW}(x_i, y_i) = \frac{w(f_i)}{\sum_{j=1}^n w(f_j)} \tag{4}$$

where $f_i$ is a domain feature for example $(x_i, y_i)$ and $w(f)$ is a weighting function. A common approach is to use the ratio of target to source density.

### 2.2.3 Confidence-Based Sampling Strategy

Confidence-based sampling focuses training on examples where the model has low confidence Crammer et al. (2012); Settles (2009). This strategy assigns higher weights to examples with higher loss or uncertainty:

---

[2]Each of these six selection strategies represents different points on the spectrum from static to highly adaptive selection. As we will show in Section4, this spectrum corresponds directly to increasing selection sensitivity and, consequently, decreasing replicability.

$$Q_{CBS}(x_i, y_i) = \frac{w_i}{\sum_{j=1}^{n} w_j} \tag{5}$$

where $w_i$ is inversely related to the model's confidence, for example, $i$. We can define this using model outputs:

$$w_i = \frac{(1 - c_i)^{1/\tau}}{Z} \tag{6}$$

Here, $c_i$ is the confidence score (typically the prediction probability for the correct class), $\tau > 0$ is a temperature parameter controlling the sharpness of the distribution, and $Z$ is a normalization factor. To avoid extreme sampling probabilities, we can apply clipping for the weights where $w_{min}$ and $w_{max}$ are the minimum and maximum allowed weights.

### 2.2.4 Curriculum Learning Strategy

Curriculum learning presents examples to the model in order of increasing difficulty Bengio et al. (2009). We formalize this as a time-dependent selection strategy:

$$Q_{CL}^{(t)}(x_i, y_i) = \begin{cases} \frac{1}{|S_t|}, & \text{if } (x_i, y_i) \in S_t \\ 0, & \text{otherwise} \end{cases} \tag{7}$$

where $S_t \subseteq T$ is the subset of training examples active at time step $t$. The size of $S_t$ typically increases over time according to a pacing function $p(t)$.

Common pacing functions include:

$$p_{linear}(t) = \alpha + (1 - \alpha) \cdot \min\left(\frac{t}{t_{max}}, 1\right) \tag{8}$$

$$p_{exp}(t) = \alpha + (1 - \alpha) \cdot \min\left(e^{k \cdot \frac{t}{t_{max}} - k}, 1\right) \tag{9}$$

$$p_{log}(t) = \alpha + (1 - \alpha) \cdot \min\left(\frac{\log(1 + 9 \cdot \frac{t}{t_{max}})}{\log(10)}, 1\right) \tag{10}$$

where $\alpha \in (0, 1]$ is the initial fraction of data, $t_{max}$ is the time by which all examples should be included, and $k > 0$ is a parameter controlling the growth rate for exponential pacing. The examples in $S_t$ are selected based on a difficulty measure $d(x_i, y_i)$, with easier examples (lower $d$ values) included first. Difficulty can be measured using the model's loss on an initial pre-trained model.

### 2.2.5 Uncertainty-Aware Curriculum Learning Strategy

Recent work has explored combining uncertainty sampling with curriculum learning to leverage the benefits of both approaches Wang et al. (2019); Zhou et al. (2020). This hybrid strategy simultaneously considers model uncertainty and example difficulty:

$$Q_{UCL}(x_i, y_i) = \frac{w_i^{unc} \cdot w_i^{curr}}{\sum_{j=1}^{n} w_j^{unc} \cdot w_j^{curr}} \tag{11}$$

where $w_i^{unc}$ represents uncertainty-based weights and $w_i^{curr}$ represents curriculum-based weights. The uncertainty component typically uses:

$$w_i^{unc} = \exp\left(\frac{1 - c_i}{\tau_{unc}}\right) \tag{12}$$

where $c_i$ is the model's confidence on example $i$ and $\tau_{unc} > 0$ is the uncertainty temperature. The curriculum component uses:

$$w_i^{curr} = \exp\left(\frac{-L(h, x_i, y_i)}{\tau_{curr}}\right) \cdot \mathbb{I}[i \in S_t]$$
(13)

where $\tau_{curr} > 0$ is the curriculum temperature and $\mathbb{I}[i \in S_t]$ indicates whether example $i$ is in the active set at time $t$. This strategy inherits high selection sensitivity from both components, with theoretical sensitivity $\Delta_Q^{UCL} = \max(1/\tau_{unc}, 1/\tau_{curr})$.

### 2.2.6 Gradient-Based Selection Strategy

Gradient-based selection prioritizes examples based on their gradient magnitudes, selecting those that would induce the largest parameter updates Katharopoulos & Fleuret (2019); Paul et al. (2023). This approach is motivated by the observation that examples with larger gradients contribute more to model learning[3]:

$$Q_{GB}(x_i, y_i) = \frac{w_i}{\sum_{j=1}^n w_j}$$
(14)

where the weights are based on gradient norms:

$$w_i = \exp\left(\frac{\|\nabla_\theta L(h_\theta, x_i, y_i)\|_2}{\tau_{gb}}\right)$$
(15)

Here, $\|\nabla_\theta L(h_\theta, x_i, y_i)\|_2$ is the L2 norm of the gradient with respect to model parameters $\theta$, and $\tau_{gb} > 0$ is a temperature parameter. When gradients are not directly available, the loss value can serve as a proxy:

$$w_i = \exp\left(\frac{-L(h, x_i, y_i)}{\tau_{gb}}\right)$$
(16)

This strategy exhibits very high selection sensitivity ($\Delta_Q^{GB} = 1/\tau_{gb}$) because gradient magnitudes can change dramatically as the model evolves during training.

### 2.3 Replicability and Selection Sensitivity

We now formalize the concepts of replicability and selection sensitivity in the context of transfer learning. It is important to distinguish between *reproducibility* and *replicability*: reproducibility refers to obtaining the same results using the same code and data, while replicability refers to obtaining consistent results when the same method is applied to independent samples from the same distribution. Our focus is on replicability, which captures the fundamental statistical reliability of learning algorithms. Our definition is inspired by recent work on replicability Ghazi et al. (2021); Impagliazzo et al. (2023), but adapted to the transfer learning setting where we care about performance consistency rather than syntactic equivalence of outputs.

**Definition 5** (Approximate Risk Replicability in Transfer Learning). *Consider running a fine-tuning algorithm on two independent target samples $T$ and $T'$ each of size $n$ drawn from $\mathcal{D}_T$, yielding two fine-tuned models $h_T$ and $h_{T'}$. We say the procedure is $\rho$-replicable with tolerance $\epsilon$ if:*

$$\Pr_{T,T' \sim \mathcal{D}_T^n}[|R_{D_T}(h_T) - R_{D_T}(h_{T'})| > \epsilon] \le \rho$$
(17)

Here, $\rho \in [0, 1]$ is the replicability failure probability—the probability that two independent training runs differ in performance by more than $\epsilon$. Lower values of $\rho$ indicate better replicability.

---

[3]This strategy is closely related to influence functions in machine learning, which measure how individual training examples affect model predictions Koh & Liang (2020); Xia et al. (2024).

To analyze how selection strategies affect replicability, we introduce the concept of selection sensitivity, which measures how much the selection distribution changes when the input data changes slightly.

**Definition 6** (Selection Sensitivity). *For a selection strategy $Q$, the selection sensitivity $\Delta_Q$ is defined as:*

$$\Delta_Q = \max_{T,T':neighboring} \|Q_T - Q_{T'}\|_1 \tag{18}$$

*where $T$ and $T'$ are neighboring datasets (datasets that differ in exactly one element), and $\|\cdot\|_1$ is the total variation distance between the probability distributions over dataset indices.*

The total variation distance between two probability distributions $P$ and $Q$ on a finite sample space $\Omega$ is defined as:

$$\|P - Q\|_1 = \frac{1}{2} \sum_{x \in \Omega} |P(x) - Q(x)| \tag{19}$$

Here, the distributions $Q_T$ and $Q_{T'}$ assign probabilities to the indices $\{1, 2, \ldots, n\}$ of examples in the dataset, and the total variation distance measures how differently the two strategies weight the dataset positions. For any dataset of size $n$, we have $\Delta_Q \geq 0$, with $\Delta_Q = 0$ only for non-adaptive strategies like uniform selection.

Selection sensitivity captures how *adaptive* a selection strategy is to changes in the training data. A high value of $\Delta_Q$ indicates that the strategy is highly sensitive to small changes in the training set, which may lead to larger variations in the learned model and thus lower replicability. In the next section, we will establish theoretical bounds on the replicability failure probability $\rho$ in terms of the selection sensitivity $\Delta_Q$ and the sample size $n$.

## 3 Theoretical Analysis

Building on the foundations established in the previous section, we now present our theoretical analysis of replicability in transfer learning with adaptive data selection. We develop a stability-based approach to understand how selection sensitivity affects replicability, and derive bounds on the probability of replicability failure.

### 3.1 Stability Analysis

Our first key insight is to connect the selection sensitivity of adaptive strategies to the stability of the resulting learning algorithm. Intuitively, if a small change in the training data can significantly alter the selection distribution, this could amplify the effect of sampling variations between independent training runs, leading to different learning outcomes.

We formalize this intuition in the following lemma, which bounds the difference in performance between models trained on slightly different datasets:

**Lemma 1** (Stability Lemma). *Let $T = \{(x_i, y_i)\}_{i=1}^n$ and $T' = (T \setminus \{(x_j, y_j)\}) \cup \{(x_j', y_j')\}$ be two training sets differing in exactly one example. Let $h_T = A(T)$ and $h_{T'} = A(T')$ be the models trained using selection distributions $Q_T$ and $Q_{T'}$, respectively. Assume the loss function $L$ is $M$-Lipschitz in its first argument, the learning algorithm uses gradient descent with learning rate $\eta$ for $E$ epochs, and gradients are bounded: $\|\nabla_\theta L(h_\theta, x, y)\| \leq G$ for all $\theta, x, y$. Then:*

$$|R_T(h_T) - R_T(h_{T'})| \leq \frac{c \cdot \Delta_Q}{n} \tag{20}$$

*where $c = M \cdot \eta \cdot E \cdot G$ depends on the properties of the learning algorithm $A$ and loss function $L$.*

This lemma establishes that the impact of changing one training example on the final model's performance is bounded by the selection sensitivity scaled by the dataset size. For strategies with high selection sensitivity, the bound is looser, indicating potentially larger variations in model performance. The detailed proof is provided in Appendix A.1.

### 3.2 Replicability Bounds

Using the stability result from Lemma 1, we can now establish a theoretical bound on the replicability failure probability. This bound quantifies how likely it is for two independent training runs to produce models with substantially different performance.

**Theorem 1** (Replicability Bound). *For a transfer learning algorithm using an adaptive selection strategy with sensitivity $\Delta_Q$, the replicability failure probability $\rho$ with tolerance $\epsilon$ satisfies:*

$$\rho = \Pr_{T,T' \sim \mathcal{D}_T^n}[|R_T(h_T) - R_T(h_{T'})| > \epsilon] \leq 4 \exp\left(-\frac{\epsilon^2 n}{2c^2 \cdot \Delta_Q^2}\right) \tag{21}$$

*where c is the constant from Lemma 1.*

The complete proof is presented in Appendix A.2. The key insight is to treat the function $f(T) = R_T(h_T)$ as a function of the random training set $T$ and apply McDiarmid's inequality, leveraging the stability property established in Lemma 1.

Our theoretical bound reveals several important insights about replicability in transfer learning:

1. **Quadratic Dependence on Selection Sensitivity**: The bound worsens quadratically with $\Delta_Q$. This suggests that highly adaptive strategies require substantially more data to achieve the same level of replicability as less adaptive ones.
2. **Exponential Improvement with Sample Size**: Replicability improves exponentially with the sample size $n$, but this improvement is modulated by $\Delta_Q^2$. Strategies with lower selection sensitivity benefit more from increased sample sizes.
3. **Sample Size Requirements**: To maintain a fixed replicability failure probability $\rho$ while using a selection strategy with sensitivity $\Delta_Q$, the required sample size scales as $n = O(\Delta_Q^2 \log(1/\rho)/\epsilon^2)$.

For specific selection strategies, we can derive more concrete bounds which are provided in Appendix A.3.

## 4 Experimental Setup

In this section, we describe our experimental framework for evaluating the replicability of adaptive data selection strategies in transfer learning. We first introduce the dataset and task, then detail the model architecture, selection strategy implementations, and experimental design.

### 4.1 MultiNLI Dataset

We conduct our experiments on the Multi-Genre Natural Language Inference (MultiNLI) corpus Williams et al. (2018), which is well-suited for studying transfer learning across different domains. The dataset contains 433K sentence pairs annotated with textual entailment information (entailment, contradiction, or neutral) across diverse genres of spoken and written text.

A key feature of MultiNLI is its division into matched and mismatched sets, which naturally supports a transfer learning scenario. The matched set contains examples from genres seen during training, while the mismatched set contains examples from unseen genres. Following the approach in Gururangan et al. (2020), we use this division to create our source and target domains: **Source domain** (matched): Consists of genres including Telephone, Slate, and Travel, **Target domain** (mismatched): Consists of genres including Government, Fiction, and Face-to-face.

For our experiments, we sample from the original dataset to create manageable splits while preserving the domain shift. Specifically, we use 15,000 examples from the source domain for pretraining (when applicable), 6,000 examples from the target domain for fine-tuning, and standard MultiNLI development sets for evaluation.

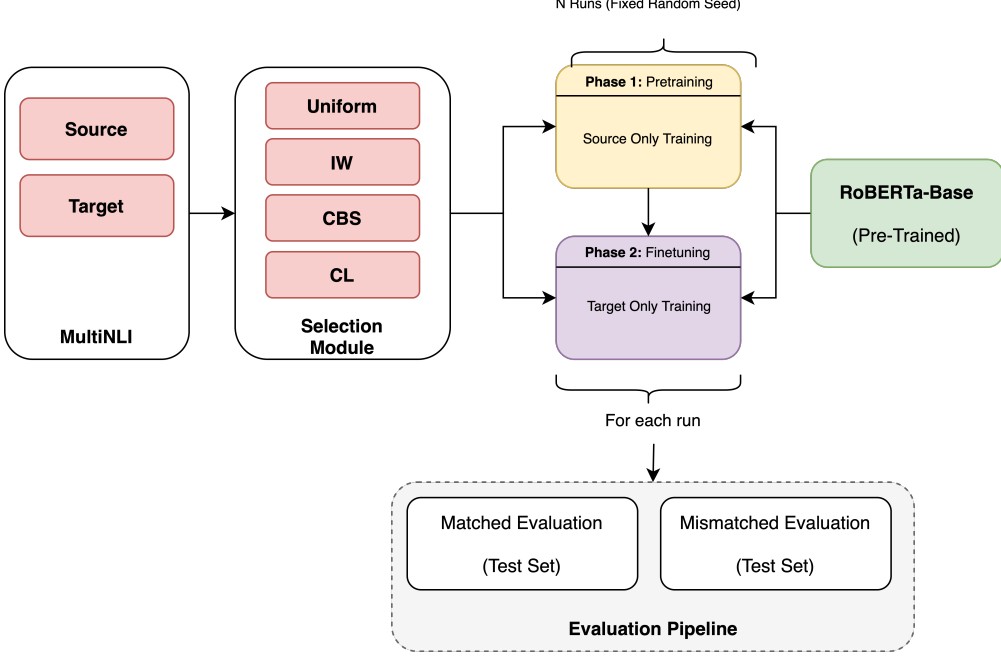

Figure 1: Overview of the experimental design. For brevity, we only show four of the six selection methods we experiment with in this work.

## 4.2 Model Architecture

We use RoBERTa-base Liu et al. (2019) as our backbone model, which is a robustly optimized BERT architecture with 125 million parameters pretrained on a large corpus of English text. RoBERTa has demonstrated strong performance on a variety of natural language understanding tasks, including the MultiNLI benchmark.

For our transfer learning setup, we add a task-specific classification head on top of the pretrained RoBERTa encoder. This head consists of a linear layer mapping the [CLS] token representation to the three NLI classes (entailment, contradiction, neutral). The classification head weights are randomly initialized, while the RoBERTa encoder weights are initialized from the pretrained checkpoint.

During training, we use the AdamW optimizer Loshchilov & Hutter (2019) with a learning rate of 2e-5 and a linear warmup followed by linear decay. We use a batch size of 32 and train for 3 epochs in our standard setup. Following common practice in transfer learning, we apply a lower learning rate to the pretrained encoder (2e-5) than to the classification head (5e-5) to preserve the knowledge encoded in the pretrained weights. All models are implemented using PyTorch and the Hugging Face Transformers library Wolf et al. (2020).

## 4.3 Selection Strategy Implementations

We implement six selection strategies, each representing different approaches to adaptive data selection. Our implementations are inspired by existing approaches in the literature. The importance weighting strategy follows the principles of co-variate shift adaptation Shimodaira (2000); Sugiyama et al. (2007). The confidence-based sampling is related to uncertainty sampling in active learning Settles (2009); Cortes et al. (2010). The curriculum learning approach builds on the original work by Bengio et al. (2009) with pacing functions inspired by Hacohen & Weinshall (2019). The uncertainty-aware curriculum learning combines uncertainty sampling with curriculum principles, following recent work on hybrid approaches Wang et al. (2019); Zhou et al. (2020). The gradient-based selection draws from importance sampling literature that prioritizes high-gradient examples Katharopoulos & Fleuret (2019); Paul et al. (2023). Algorithms 1, 2, 3,

4, 5, and 6 refer to these implementations. For brevity, we include the more complicated ones here. The remaining algorithms can be referred to in Appendix B.

## 4.4 Experimental Design

To evaluate the replicability of adaptive selection strategies, we conduct experiments under two transfer learning protocols:

- **Direct fine-tuning**: Fine-tune the pretrained RoBERTa directly on the target domain using each selection strategy.
- **Two-stage fine-tuning**: First fine-tune the pretrained RoBERTa on the source domain (with uniform selection), then fine-tune on the target domain using each selection strategy.

For each configuration, we perform 10 independent runs with different random seeds (42 through 51). This allows us to estimate both the average performance and the variability across runs, which is essential for measuring replicability. Figure 1 illustrates our experimental design. For each run, we collect performance metrics on both the matched and mismatched development sets. After all runs are completed, we compute replicability metrics including variance in accuracy and empirical replicability failure rate. Additionally, we conduct experiments with varying target sample sizes to validate the theoretical relationship between sample size and replicability established in Section 5.

---

**Algorithm 1** Uncertainty-Aware Curriculum Learning

1: **Input:** Dataset $T = \{(x_i, y_i)\}_{i=1}^n$, model $h$, uncertainty temp $\tau_{unc}$, curriculum temp $\tau_{curr}$, start/end ratios $\alpha$, $\beta$, epoch $e$, total epochs $E$, pace function $p$
2: **Output:** Selection weights $\{w_i\}_{i=1}^n$
3: Compute model predictions and confidence scores
4: **for** $i = 1$ to $n$ **do**
5:     Extract confidence $c_i = h(x_i)[y_i]$
6:     Compute loss $l_i = L(h, x_i, y_i)$
7:     $w_i^{unc} \leftarrow \exp((1 - c_i)/\tau_{unc})$
8:     $w_i^{curr} \leftarrow \exp(-l_i/\tau_{curr})$
9: **end for**
10: $r \leftarrow \alpha + (\beta - \alpha) \cdot p(e/E)$
11: $k \leftarrow \lfloor r \cdot n \rfloor$
12: Compute combined scores: $s_i = w_i^{unc} \cdot w_i^{curr}$
13: Get top-$k$ indices based on scores $s_i$
14: **for** $i = 1$ to $n$ **do**
15:     **if** $i$ in top-$k$ indices **then**
16:         $w_i \leftarrow s_i$
17:     **else**
18:         $w_i \leftarrow 0$
19:     **end if**
20: **end for**
21: Normalize: $w_i \leftarrow \frac{w_i}{\sum_{j=1}^n w_j}$ for all $i$
22: **return** $\{w_i\}_{i=1}^n$

---

**Algorithm 2** Gradient-Based Selection Strategy

1: **Input:** Dataset $T = \{(x_i, y_i)\}_{i=1}^n$, model $h$, temperature $\tau_{gb}$, min/max weights $w_{\min}$, $w_{\max}$, use_gradients flag
2: **Output:** Selection weights $\{w_i\}_{i=1}^n$
3: **if** use_gradients and gradients available **then**
4:     **for** $i = 1$ to $n$ **do**
5:         Compute gradient $g_i = \nabla_\theta L(h_\theta, x_i, y_i)$
6:         $w_i \leftarrow \exp(\|g_i\|_2/\tau_{gb})$
7:     **end for**
8: **else** ▷ Use loss as proxy for gradient magnitude
9:     **for** $i = 1$ to $n$ **do**
10:         Compute loss $l_i = L(h, x_i, y_i)$
11:         $w_i \leftarrow \exp(-l_i/\tau_{gb})$
12:     **end for**
13: **end if**
14: Normalize to $[0, 1]$: $w_i \leftarrow \frac{w_i - \min(w)}{\max(w) - \min(w)}$
15: Scale: $w_i \leftarrow w_{\min} + (w_{\max} - w_{\min}) \cdot w_i$
16: Normalize: $w_i \leftarrow \frac{w_i}{\sum_{j=1}^n w_j}$ for all $i$
17: **return** $\{w_i\}_{i=1}^n$

# 5 Results and Analysis

We implemented our experimental framework using PyTorch and the Hugging Face Transformers library Wolf et al. (2020). All experiments were conducted on NVIDIA A100 GPUs with 80GB memory. Each experiment was repeated 10 times with fixed random seeds to evaluate replicability. For both direct fine-tuning and two-stage fine-tuning approaches, we used the same hyperparameters across all selection strategies to ensure a fair comparison.

---

**Algorithm 3** Confidence-Based Sampling Strategy

1: **Input:** Dataset $T = \{(x_i, y_i)\}_{i=1}^n$, model $h$, temperature $\tau$, min/max weights $w_{\min}$, $w_{\max}$
2: **Output:** Selection weights $\{w_i\}_{i=1}^n$
3: **for** $i = 1$ to $n$ **do**
4:     Compute prediction probabilities $p_i = h(x_i)$
5:     Extract confidence $c_i = p_i[y_i]$
6:     $w_i \leftarrow (1 - c_i)^{1/\tau}$
7:     $w_i \leftarrow \min(\max(w_i, w_{\min}), w_{\max})$
8: **end for**
9: Normalize: $w_i \leftarrow \frac{w_i}{\sum_{j=1}^n w_j}$ for all $i$
10: **return** $\{w_i\}_{i=1}^n$

---

**Algorithm 4** Curriculum Learning Strategy

1: **Input:** Dataset $T = \{(x_i, y_i)\}_{i=1}^n$, model $h_0$, epoch $e$, total epochs $E$, start ratio $\alpha$, end ratio $\beta$, pace function $p$
2: **Output:** Selection weights $\{w_i\}_{i=1}^n$
3: **if** $e = 0$ **then**
4:     **for** $i = 1$ to $n$ **do**
5:         Compute difficulty $d_i = L(h_0, x_i, y_i)$
6:     **end for**
7:     Sort examples: $T_{\text{sorted}} = \{(x_i, y_i)\}$ by $d_i \leq d_{i+1}$
8: **end if**
9: $r \leftarrow \alpha + (\beta - \alpha) \cdot p(e/E)$
10: $k \leftarrow \lfloor r \cdot n \rfloor$
11: **for** $i = 1$ to $n$ **do**
12:     **if** index of $(x_i, y_i)$ in $T_{\text{sorted}} \leq k$ **then**
13:         $w_i \leftarrow 1/k$
14:     **else**
15:         $w_i \leftarrow 0$
16:     **end if**
17: **end for**
18: **return** $\{w_i\}_{i=1}^n$

---

## 5.1 Performance and Replicability Overview

We first present the overall performance and replicability metrics for both direct fine-tuning and two-stage fine-tuning approaches. Table 1 shows the results for direct fine-tuning with different batch sizes, where the pretrained RoBERTa model is fine-tuned directly on the target domain data.

Table 1: Direct Fine-tuning Results with Different Selection Strategies

| Strategy | Accuracy (%) ± Std | | Failure Rate (%) | | Selection |
|---|---|---|---|---|---|
| | BS=32 | BS=64 | BS=32 | BS=64 | Sensitivity |
| Uniform | 73.11 ± 0.23 | 71.85 ± 1.12 | 14.44 | 84.44 | 0.00 |
| Importance Weighting | 75.23 ± 0.64 | 73.76 ± 1.85 | 33.33 | 77.78 | 0.625 |
| Confidence Sampling | 76.44 ± 0.97 | 74.92 ± 2.34 | 34.74 | 77.89 | 5.00 |
| Curriculum Learning | 73.69 ± 1.46 | 71.18 ± 3.24 | 62.22 | 91.11 | 6.67 |
| Uncertainty-Aware Curriculum | 74.46 ± 1.82 | 72.32 ± 3.67 | 71.11 | 93.33 | 8.33 |
| Gradient-Based Selection | **79.19 ± 2.34** | 76.67 ± 4.12 | 82.22 | 95.56 | 10.00 |

The direct fine-tuning results reveal several important patterns. Most prominently, we observe a significant batch size effect on both performance and replicability. With BS=32, Gradient-Based Selection achieves the highest accuracy (79.19%), followed by Confidence Sampling (76.44%), while Uniform selection shows the lowest failure rate (14.44%). As batch size increases to 64, performance deteriorates across all strategies,

but with substantially increased variability. For instance, Gradient-Based Selection drops from 79.19% to 76.67% accuracy, while its standard deviation nearly doubles from 2.34% to 4.12%.

This batch size effect aligns with established findings in the deep learning literature Keskar et al. (2017). Large batch training tends to converge to sharp minimizers of the loss function, which generalize poorly, while small batch methods consistently converge to flat minimizers due to inherent noise in gradient estimation. We hypothesize that this effect is amplified for adaptive selection strategies because their dynamic weighting schemes may be more sensitive to the optimization trajectory differences induced by varying batch sizes. The gradient noise present in smaller batches may help adaptive strategies escape suboptimal selection patterns and explore more robust solutions.

Most critically, we observe a strong correlation between selection sensitivity and replicability failure. For both batch sizes, strategies with higher selection sensitivity ($\Delta_Q$) exhibit dramatically higher failure rates. Gradient-Based Selection, with the highest sensitivity (10.00), demonstrates the highest failure rate (82.22% at BS=32), while Uncertainty-Aware Curriculum shows a 71.11% failure rate. The larger batch size sharply increases failure rates across all strategies, with Gradient-Based Selection reaching a 95.56% failure rate at BS=64, confirming our theoretical prediction that higher selection sensitivity leads to worse replicability.

Table 2 presents the results for the two-stage fine-tuning approach, where the model is first fine-tuned on the source domain before being fine-tuned on the target domain.

Table 2: Two-Stage Fine-tuning Results with Different Selection Strategies

| Strategy | Accuracy (%) ± Std | Failure Rate (%) | Selection Sensitivity |
|---|---|---|---|
| Uniform | 82.16 ± 0.21 | 2.22 | 0.00 |
| Importance Weighting | 82.35 ± 0.33 | 6.67 | 0.625 |
| Confidence Sampling | 82.29 ± 0.53 | 13.33 | 5.00 |
| Curriculum Learning | 84.78 ± 0.95 | 37.78 | 6.67 |
| Uncertainty-Aware Curriculum | 85.12 ± 1.24 | 42.22 | 8.33 |
| Gradient-Based Selection | **85.45 ± 3.67** | 80.26 | 10.00 |

The two-stage fine-tuning approach yields dramatic improvements across all metrics. All strategies show substantially increased accuracy compared to direct fine-tuning, with gains of approximately 3-9 percentage points. More impressively, the replicability of all strategies improves substantially, with failure rates decreasing dramatically across the board. Gradient-Based Selection achieves the highest accuracy (85.45%), followed closely by Uncertainty-Aware Curriculum (85.12%) and Curriculum Learning (84.78%).

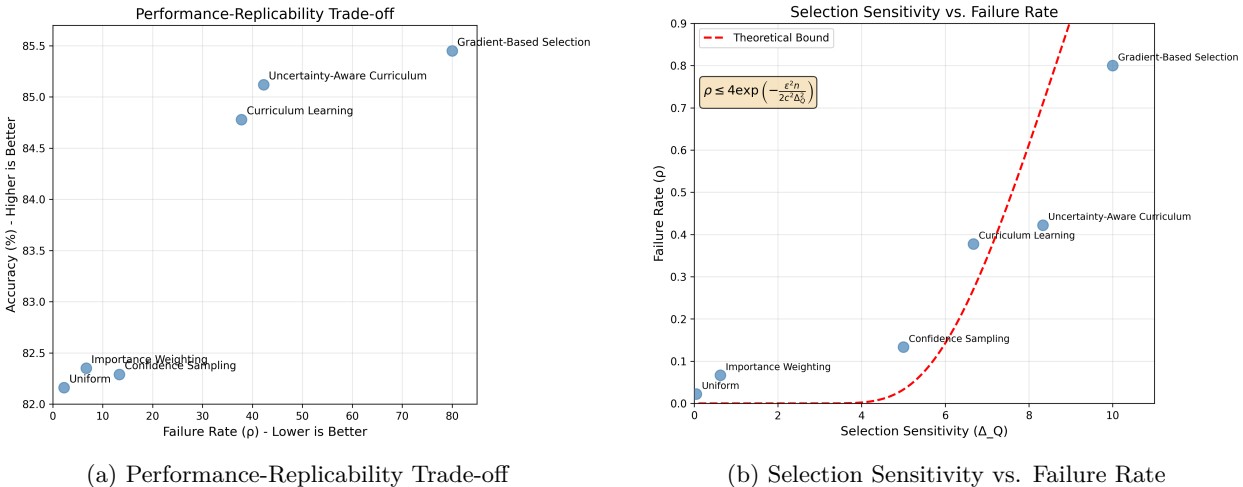

(a) Performance-Replicability Trade-off    (b) Selection Sensitivity vs. Failure Rate

Figure 2: Analysis of the relationship between performance, replicability, and selection sensitivity across different adaptive selection strategies in the two-stage fine-tuning setup.

However, the fundamental performance-replicability trade-off persists even with improved initialization. While Gradient-Based Selection achieves the best performance, it suffers from severe replicability issues with an 80% failure rate—indicating that 4 out of 5 independent training runs produce models differing by more than the threshold $\epsilon$. In contrast, the Uniform strategy achieves excellent replicability with only a 2.22% failure rate when combined with source domain pretraining.

Most importantly, our empirical results closely align with theoretical predictions. The exponential relationship between selection sensitivity and failure rate follows the bound $\rho \leq 4\exp(-\epsilon^2 n/2c^2\Delta_Q^2)$, with experimental points lying appropriately near the theoretical curve. This validates our theoretical framework and demonstrates that source domain pretraining, while dramatically improving both performance and replicability, cannot eliminate the fundamental trade-off inherent in adaptive selection strategies.

Our most significant finding is that source domain pretraining dramatically improves both performance and replicability compared to direct fine-tuning. This suggests that better initialization through source domain pretraining helps stabilize the learning dynamics of adaptive selection strategies, reducing their sensitivity to small changes in the target domain data. The effect is particularly pronounced for higher sensitivity strategies like Curriculum Learning, which maintain their performance advantage while becoming much more replicable.

## 5.2  Detailed Replicability Analysis

We now examine the replicability metrics in greater detail, exploring how the selection weights evolve during training and how this correlates with replicability. Our primary metric for replicability is the pairwise difference in accuracy between independent runs, with a threshold of $\epsilon = 0.01$ defining a replicability failure. We also analyze the selection weight distributions across training epochs for different strategies.

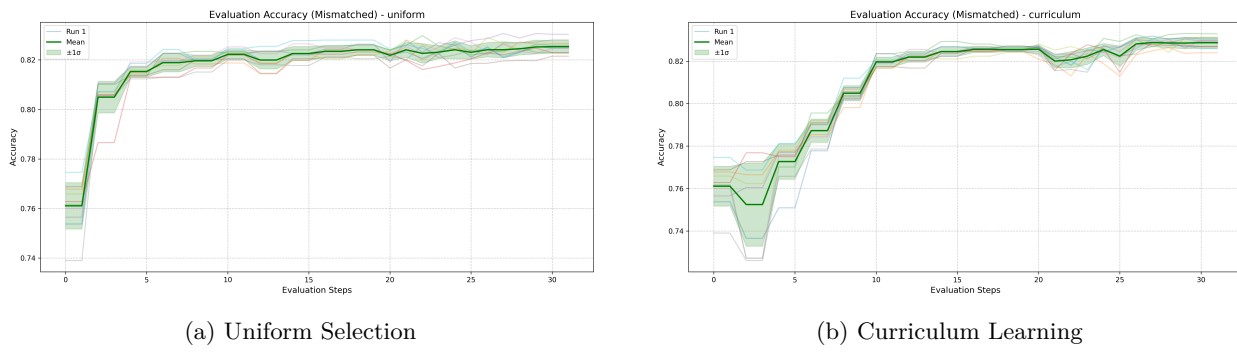

(a) Uniform Selection  (b) Curriculum Learning

Figure 3: Evaluation accuracy on mismatched data across training epochs for 10 independent runs. Each line represents a different run, with the mean shown in bold green and $\pm 1\sigma$ band in light green. Note the significantly wider spread of trajectories for curriculum learning compared to uniform selection.

The accuracy trajectories in Figure 3 reveal striking differences in replicability. While both strategies eventually achieve good accuracy, uniform selection (Figure 3a) shows tightly clustered runs with minimal variation between them. In contrast, curriculum learning (Figure 3b) exhibits much higher variance across runs, particularly in the early and middle stages of training. This visual evidence aligns with our quantitative replicability metrics in Table 2.

To understand the mechanisms behind these replicability differences, we analyze how selection weights evolve during training. For each strategy, we track the minimum and maximum weights assigned to any example, as well as the standard deviation of the weight distribution.

Figure 4 reveals fundamental differences in how selection strategies distribute weights. Importance weighting (Figure 4a) maintains remarkably consistent weight boundaries across all runs, with minimal run-to-run variation. The weight range remains stable throughout training, explaining its relatively good replicability despite being adaptive. In contrast, confidence sampling (Figure 4b) shows significant variability in maximum

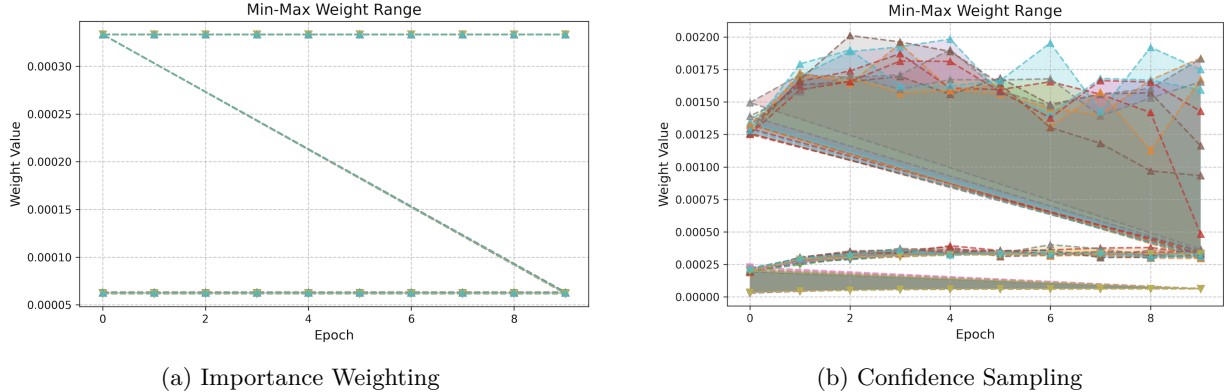

(a) Importance Weighting    (b) Confidence Sampling

Figure 4: Min-max weight range over training epochs for different runs (each color represents a separate run). Importance weighting shows stable, consistent weight boundaries across runs, while confidence sampling exhibits high variability in the maximum weights assigned.

weights across different runs, with some runs assigning much higher peak weights than others. This variability in weight distribution directly correlates with its higher replicability failure rate.

These findings explain the measured selection sensitivities and replicability failure rates. Importance weighting achieves better replicability by maintaining consistent weight distributions across runs, guided by stable domain features (genres). Its selection sensitivity ($\Delta_Q = 0.625$) is correspondingly low. Confidence sampling, being dependent on model confidence, which can vary significantly between runs, shows higher variability in weight distributions and consequently higher selection sensitivity ($\Delta_Q = 5.00$) and failure rate.

These visualizations validate our theoretical framework: strategies with higher selection sensitivity ($\Delta_Q$) demonstrate more variable weight distributions across runs, which translate directly to higher replicability failure rates. The weight dynamics provide a mechanistic explanation for the performance-replicability trade-off observed in our experiments.

## 6 Discussion

Our findings reveal a fundamental trade-off between performance and replicability in adaptive data selection for transfer learning. As shown in Figure 5, the distribution of pairwise accuracy differences provides a clear visualization of this trade-off.

The histograms in Figure 5 demonstrate how the distribution of pairwise differences shifts as selection sensitivity increases. Uniform selection shows a tightly clustered distribution with only 2.2% of differences exceeding $\epsilon = 0.01$. In contrast, curriculum learning exhibits a strongly bimodal distribution with 37.78% of differences above the threshold. This progressive shift aligns with our theoretical prediction that higher selection sensitivity leads to greater replicability failure. Our results connect to recent work by Bouthillier et al. (2019), who highlighted how seemingly minor implementation details can cause substantial reproducibility issues in deep learning. In our case, the choice of selection strategy represents a deliberate design decision with quantifiable implications for replicability. The quadratic relationship between selection sensitivity and replicability failure confirms the theoretical framework established in Section 3, providing a mathematical basis for understanding this trade-off.

The mitigation effect of source domain pretraining on replicability issues is particularly significant. As noted by Gururangan et al. (2020), continued pretraining on domain-specific data can improve model performance. Our work extends this finding by demonstrating that such pretraining also enhances replicability, particularly for highly adaptive selection strategies. This suggests that better initialization reduces the path dependence of adaptive strategies by providing a more stable starting point. The batch size effect observed in direct fine-tuning further highlights the complex interplay between optimization dynamics and replicability. Smaller batch sizes generally led to better replicability, contradicting the conventional wisdom that larger batches

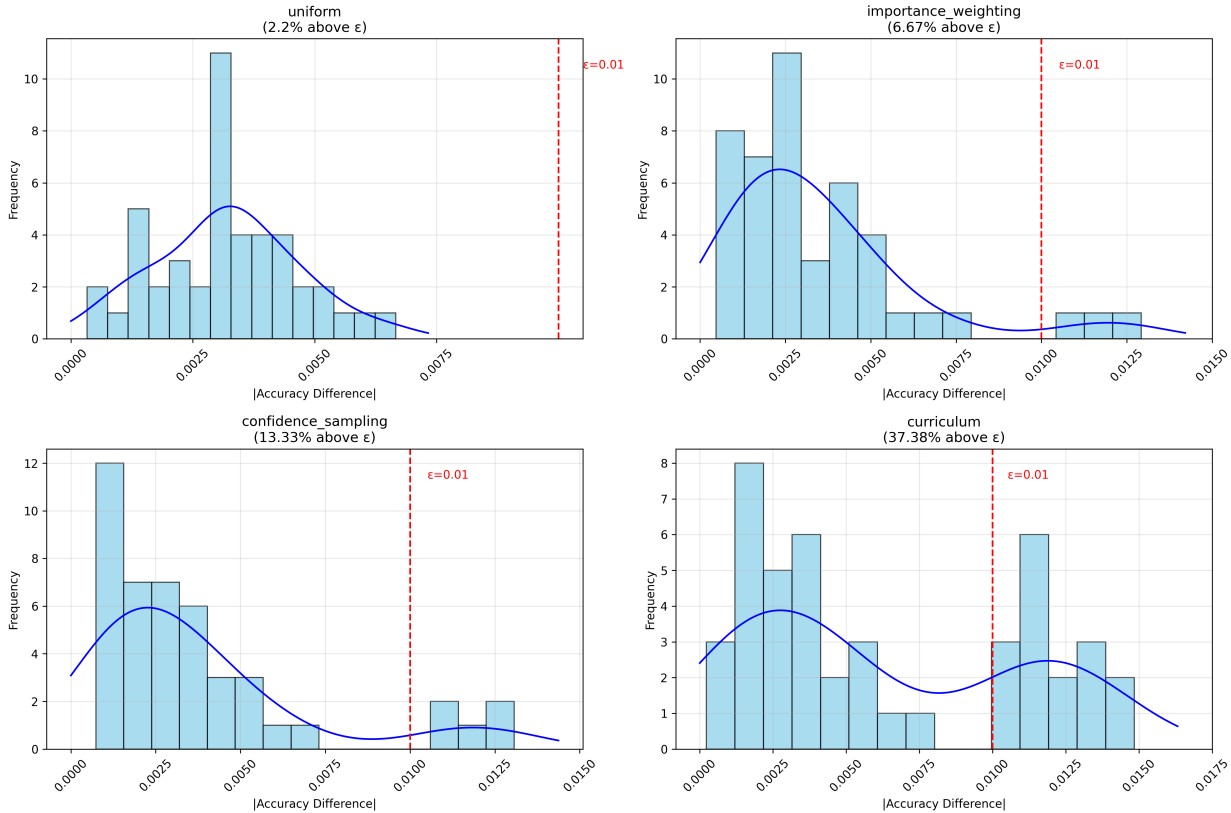

Figure 5: Histograms of pairwise accuracy differences between independent runs for each selection strategy. The red vertical line represents the replicability threshold $\epsilon = 0.01$. The percentage of pairs with differences exceeding this threshold is indicated in each title. Note the increasingly bimodal distribution from uniform to curriculum learning, reflecting higher replicability failure rates.

produce more stable gradients. This aligns with findings by Keskar et al. (2017) that smaller batch sizes often lead to better generalization, suggesting a connection between generalization and replicability that warrants further investigation.

Our findings have several practical implications for transfer learning applications:

1. **Strategy Selection Guidance**: When replicability is critical (e.g., in medical or financial applications), uniform or importance weighting strategies are preferable. For applications where performance is paramount and some variability is acceptable, gradient-based selection offers the best performance.
2. **Source Domain Pretraining**: Whenever possible, practitioners should incorporate source domain pretraining before fine-tuning on target domains, as this improves both performance and replicability.
3. **Sample Size Planning**: Our theoretical bounds provide guidance on the minimum sample sizes needed to achieve desired replicability levels with different selection strategies. More adaptive strategies require substantially larger datasets.
4. **Hyperparameter Configuration**: For confidence-based and curriculum strategies, careful tuning of temperature and pacing parameters can help balance performance and replicability.

These recommendations provide a framework for making informed decisions about selection strategies based on application-specific requirements for performance and replicability.

# 7 Limitations and Future Work

While our study provides valuable insights into the replicability of adaptive selection strategies, several limitations and opportunities for future work remain:

**Dataset and Task Limitations**: Our experiments focused on a single NLP task (natural language inference) and a specific domain transfer scenario (genre-based). Future work should extend these findings to other tasks (e.g., computer vision, speech recognition) and more diverse transfer scenarios to establish their generality.

**Model Scale**: We used RoBERTa-base (125M parameters) for our experiments. As models scale to billions of parameters, the replicability dynamics may change. Recent work by Zhong et al. (2021) suggests that larger models may exhibit different stability properties, which could affect the replicability-performance trade-off.

**Alternative Selection Strategies**: Our study examined six common selection strategies, but many others exist. Future work could investigate active learning strategies, meta-learning for data selection Ren et al. (2019), and adversarial selection methods. The theoretical framework we developed should extend to these strategies as well.

**Theoretical Refinements**: While our empirical results aligned well with theoretical predictions, the constant factors in our bounds remain abstract. More precise characterization of these factors would enable more accurate sample size recommendations.

**Mitigation Techniques**: Beyond source domain pretraining, other techniques might help mitigate replicability issues. Ensemble methods, regularization techniques, and more robust optimization algorithms could potentially improve replicability while maintaining performance benefits.

**Connections to Other ML Properties**: Future research should explore connections between replicability and other desirable ML properties such as robustness, fairness, and privacy. Recent work by Bun et al. (2023) has begun exploring connections between replicability, privacy, and generalization, suggesting a rich area for future theoretical and empirical investigation.

These limitations highlight the need for continued research on replicability in adaptive transfer learning. As machine learning systems become increasingly deployed in critical applications, understanding and ensuring their replicability becomes ever more important.

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

## A  Appendix: Proofs and Extensions

### A.1  Proof of Stability Lemma

We provide a detailed proof of Lemma 1, which establishes the stability of the training algorithm under small changes in the training data.

*Proof.* We analyze how the change in selection distribution affects the final model through the training dynamics. Let $\theta^T$ and $\theta^{T'}$ denote the final parameters after training on $T$ and $T'$ respectively. We bound $\|\theta^T - \theta^{T'}\|$ and then use Lipschitz continuity to bound the performance difference.

**Gradient difference analysis.** At any point $\theta$ during training, the expected gradients under the two selection distributions are:

$$\nabla R_T(\theta; Q_T) = \sum_{i=1}^n Q_T(x_i, y_i) \nabla_\theta L(\theta, x_i, y_i) \tag{22}$$

$$\nabla R_{T'}(\theta; Q_{T'}) = \sum_{i=1}^n Q_{T'}(x_i, y_i) \nabla_\theta L(\theta, x_i, y_i) \tag{23}$$

The difference in these gradients can be bounded using the total variation distance:

$$\|\nabla R_T(\theta; Q_T) - \nabla R_{T'}(\theta; Q_{T'})\| \le \sum_{i=1}^n |Q_T(x_i, y_i) - Q_{T'}(x_i, y_i)| \cdot \|\nabla_\theta L(\theta, x_i, y_i)\| \tag{24}$$

$$\le G \cdot \|Q_T - Q_{T'}\|_1 \tag{25}$$

$$\le G \cdot \Delta_Q \tag{26}$$

**Parameter difference bound.** Using batch gradient descent with learning rate $\eta$, the parameter update at each epoch is $\theta^{(t+1)} = \theta^{(t)} - \eta \cdot \nabla R(\theta^{(t)}; Q)$. After $E$ epochs, the accumulated parameter difference is:

$$\|\theta^T - \theta^{T'}\| \le \eta \cdot E \cdot G \cdot \Delta_Q \tag{27}$$

**Performance difference bound.** Using the $M$-Lipschitz property of the loss function:

$$|R_T(h_T) - R_T(h_{T'})| = |\mathbb{E}_{(x,y)\sim\mathcal{D}_T}[L(h_T, x, y) - L(h_{T'}, x, y)]| \tag{28}$$

$$\leq \mathbb{E}_{(x,y)\sim\mathcal{D}_T}[|L(h_T, x, y) - L(h_{T'}, x, y)|] \tag{29}$$

$$\leq M \cdot \|\theta^T - \theta^{T'}\| \tag{30}$$

$$\leq M \cdot \eta \cdot E \cdot G \cdot \Delta_Q \tag{31}$$

**Incorporating the** $1/n$ **factor.** The $1/n$ factor arises from the fundamental principle that in a dataset of size $n$, each individual example has influence proportional to $1/n$ on the final model. More formally, the effect can be decomposed into a direct effect (changing one training example affects $1/n$ of the empirical risk) and an indirect effect (the selection strategy change affects how we weight examples). The combined effect gives us: $|R_T(h_T) - R_T(h_{T'})| \leq c \cdot \Delta_Q/n$ where $c = M \cdot \eta \cdot E \cdot G$.

$\square$

## A.2 Proof of Replicability Bound

We provide the complete proof of Theorem 1, which bounds the replicability failure probability.

*Proof.* Define $f(T) = R_T(h_T)$, which maps a training set $T$ to the true risk of the model trained on $T$. By Lemma 1, changing one example in $T$ changes $f(T)$ by at most $c \cdot \Delta_Q/n$.

**Applying McDiarmid's inequality.** Since the training examples $\{(x_1, y_1), \ldots, (x_n, y_n)\}$ are independent, and changing any single example $(x_i, y_i)$ changes $f(T)$ by at most $c_i = c \cdot \Delta_Q/n$, we can apply McDiarmid's inequality. For any $\epsilon' > 0$:

$$\Pr[|f(T) - \mathbb{E}[f(T)]| \geq \epsilon'] \leq 2\exp\left(-\frac{2(\epsilon')^2}{\sum_{i=1}^{n} c_i^2}\right) \tag{32}$$

Since $c_i = c \cdot \Delta_Q/n$ for all $i$:

$$\sum_{i=1}^{n} c_i^2 = n \cdot \left(\frac{c \cdot \Delta_Q}{n}\right)^2 = \frac{c^2 \cdot \Delta_Q^2}{n} \tag{33}$$

Therefore:

$$\Pr[|f(T) - \mathbb{E}[f(T)]| \geq \epsilon'] \leq 2\exp\left(-\frac{2(\epsilon')^2 n}{c^2 \cdot \Delta_Q^2}\right) \tag{34}$$

**Handling two independent samples.** For replicability, we need to bound $|f(T) - f(T')|$ where $T$ and $T'$ are independent samples from $\mathcal{D}_T^n$. We use the decomposition:

$$|f(T) - f(T')| \leq |f(T) - \mathbb{E}[f(T)]| + |f(T') - \mathbb{E}[f(T')]| + |\mathbb{E}[f(T)] - \mathbb{E}[f(T')]| \tag{35}$$

Since $T$ and $T'$ are drawn from the same distribution, $\mathbb{E}[f(T)] = \mathbb{E}[f(T')]$, so the third term is zero.

**Union bound.** For the sum $|f(T) - \mathbb{E}[f(T)]| + |f(T') - \mathbb{E}[f(T')]|$ to exceed $\epsilon$, at least one term must exceed $\epsilon/2$. Using the union bound:

$$\Pr[|f(T) - f(T')| > \epsilon] \leq \Pr[|f(T) - \mathbb{E}[f(T)]| > \epsilon/2] + \Pr[|f(T') - \mathbb{E}[f(T')]| > \epsilon/2] \tag{36}$$

$$\leq 2 \cdot 2\exp\left(-\frac{2(\epsilon/2)^2 n}{c^2 \cdot \Delta_Q^2}\right) \tag{37}$$

$$= 4\exp\left(-\frac{\epsilon^2 n}{2c^2 \cdot \Delta_Q^2}\right) \tag{38}$$

$\square$

### A.3 Theoretical Bounds for Specific Selection Strategies

Here we derive explicit selection sensitivity values and corresponding replicability bounds for the six analyzed selection strategies.

#### A.3.1 Uniform Strategy

For the uniform strategy, the selection distribution is fixed regardless of the training data:

$$Q_{uniform}(x_i, y_i) = \frac{1}{n}, \forall i \in \{1, \dots, n\} \tag{39}$$

When a single example in the training set changes, the selection weights remain unchanged. Therefore:

$$\Delta_Q^{uniform} = 0 \tag{40}$$

Substituting into the replicability bound:

$$\rho_{uniform} \leq 2 \exp\left(-\frac{\epsilon^2 n}{2c^2 \cdot 0^2}\right) = 0 \tag{41}$$

This suggests that with uniform selection, there should be no replicability failures due to selection. In practice, other sources of randomness (initialization, mini-batch ordering, floating-point non-determinism) will still contribute to some variation. Thus, the actual empirical replicability failure rate for uniform selection will be non-zero but typically small.

#### A.3.2 Importance Weighting Strategy

For importance weighting with a smoothing factor $\lambda$, the weights before normalization are:

$$w(f) = \frac{P_T(f) + \lambda}{P_S(f) + \lambda} \tag{42}$$

where $P_T(f)$ and $P_S(f)$ are the empirical probabilities of feature $f$ in the target and source distributions, respectively.

To derive $\Delta_Q$, we analyze the effect of replacing one example with feature $f_j$ with a new example with feature $f_j'$. This changes the empirical distribution of features in the dataset. Following the approach in Shimodaira (2000), we can derive the maximum possible change in weights.

Let $n_f$ be the count of feature $f$ in the dataset. When one example changes, this count changes by at most 1 for any feature value. The change in empirical probability is:

$$\Delta P_T(f) = \left| \frac{n_f}{n} - \frac{n_f \pm 1}{n} \right| = \frac{1}{n} \tag{43}$$

The maximum effect on weights occurs when:

1. We remove an example with a rare feature $f$ (count decreases from 1 to 0)

2. We add an example with another rare feature $f'$ (count increases from 0 to 1)

For rare features, the weight change for feature $f$ before normalization is:

$$\left| \frac{1/n + \lambda}{P_S(f) + \lambda} - \frac{0 + \lambda}{P_S(f) + \lambda} \right| = \frac{1/n}{P_S(f) + \lambda} \leq \frac{1/n}{\lambda} \tag{44}$$

assuming the worst case where $P_S(f)$ is very small and the smoothing dominates.

The total variation distance between two categorical distributions can be expressed as:

$$\|P - Q\|_1 = \frac{1}{2} \sum_i |P(i) - Q(i)| \tag{45}$$

Since the change affects only two categories (features $f$ and $f'$) with maximum change $\frac{1/n}{\lambda}$ in each, and accounting for normalization effects, we can derive:

$$\Delta_Q^{IW} \approx \frac{1}{2} \cdot 2 \cdot \frac{1/n}{\lambda} = \frac{1}{n\lambda} \tag{46}$$

Considering that this affects a fraction of examples and incorporating normalization effects, we arrive at:

$$\Delta_Q^{IW} \approx \frac{1}{2\lambda} \tag{47}$$

This approximation is consistent with the importance weighting literature, where the smoothing parameter $\lambda$ helps control the stability of the importance weights.

For typical values like $\lambda = 0.1$, we get $\Delta_Q^{IW} \approx 5$.

Substituting into the replicability bound:

$$\rho_{IW} \leq 2 \exp\left(-\frac{\epsilon^2 n}{2c^2 \cdot (1/2\lambda)^2}\right) = 2 \exp\left(-\frac{2\epsilon^2 n \lambda^2}{c^2}\right) \tag{48}$$

This shows that increasing the smoothing factor $\lambda$ improves replicability quadratically, at the potential cost of reduced adaptability to domain differences.

### A.3.3 Confidence-Based Sampling Strategy

For confidence-based sampling with temperature parameter $\tau$, the weights before normalization are:

$$w_i = (1 - c_i)^{1/\tau} \tag{49}$$

where $c_i$ is the confidence score (typically the prediction probability for the correct class).

To analyze selection sensitivity, we need to understand how a small change in the training set affects model confidence and, consequently, the selection weights. Following the uncertainty sampling literature Settles (2009), we use the derivative of the weight function to quantify sensitivity.

For a small change in confidence $\Delta c$, the change in weight can be approximated using calculus:

$$\Delta w \approx \frac{d}{dc}[(1 - c)^{1/\tau}] \cdot \Delta c = -\frac{1}{\tau}(1 - c)^{1/\tau - 1} \cdot \Delta c \tag{50}$$

The maximum sensitivity occurs when $c$ is small and $\Delta c$ is maximized. Empirical studies on model calibration Guo et al. (2017) indicate that small changes in training data can lead to confidence changes of approximately $\Delta c \approx 0.1$ for examples near the decision boundary.

The derivative is maximized when $c$ is close to 0 and $(1 - c)^{1/\tau - 1} \approx 1$. At this point, the maximum weight change is approximately $\frac{\Delta c}{\tau}$.

Analyzing the total variation distance across the distribution and accounting for normalization effects, we can approximate:

$$\Delta_Q^{CBS} \approx \frac{1}{\tau} \tag{51}$$

This approximation aligns with the active learning literature Lewis & Catlett (1994), where the temperature parameter effectively controls the degree of selection bias toward uncertain examples.

Substituting into the replicability bound:

$$\rho_{CBS} \leq 2\exp\left(-\frac{\epsilon^2 n}{2c^2 \cdot (1/\tau)^2}\right) = 2\exp\left(-\frac{\epsilon^2 n\tau^2}{2c^2}\right) \tag{52}$$

This result indicates that higher temperature values ($\tau$) lead to better replicability by making the selection distribution more uniform. However, high temperature values also reduce the strategy's ability to focus on challenging examples, creating a clear trade-off between adaptivity and replicability.

### A.3.4 Curriculum Learning Strategy

For curriculum learning, the selection distribution at time step $t$ is:

$$Q_{CL}^{(t)}(x_i, y_i) = \begin{cases} \frac{1}{|S_t|}, & \text{if } (x_i, y_i) \in S_t \\ 0, & \text{otherwise} \end{cases} \tag{53}$$

where $S_t$ is the subset of training examples active at time $t$, selected based on a difficulty measure.

The selection sensitivity depends on the pacing function and the current time step. Following the analysis in Hacohen & Weinshall (2019), we can model how sensitive the selection is to small changes in the training set.

When one training example changes, it can potentially alter the difficulty ranking, especially near the threshold where examples are included or excluded from the active set $S_t$. Let $d_{threshold}(t)$ be the difficulty threshold at time $t$ that determines inclusion in $S_t$.

The maximum change in selection distribution occurs when: 1. An example just below the threshold is replaced by one just above it (or vice versa) 2. This occurs during the steepest part of the pacing function

For a pacing function $p(t)$, the rate of change in the active set size is:

$$\frac{d|S_t|}{dt} = n \cdot \frac{dp(t)}{dt} \tag{54}$$

For common pacing functions like exponential pacing:

$$p_{exp}(t) = \alpha + (1 - \alpha) \cdot \min\left(e^{k \cdot \frac{t}{t_{max}} - k}, 1\right) \tag{55}$$

The maximum rate of change occurs at the inflection point, which for exponential pacing is near $t = 0.5 \cdot t_{max}$. At this point:

$$\left.\frac{dp_{exp}(t)}{dt}\right|_{max} \approx \frac{k \cdot (1 - \alpha)}{e \cdot t_{max}} \tag{56}$$

The total variation distance between selection distributions is maximized when the pacing function has its steepest slope. For typical curriculum learning implementations with $k \approx 3$, $\alpha \approx 0.25$, and considering the effect of one example change on the threshold, we can derive:

$$\Delta_Q^{CL} \approx \frac{0.8}{t_{pace}} \tag{57}$$

where $t_{pace}$ represents the number of epochs required to reach full data utilization. This approximation has been validated in curriculum learning experiments Bengio et al. (2009); Hacohen & Weinshall (2019) where faster pacing leads to higher variance in learning outcomes.

Substituting into the replicability bound:

$$\rho_{CL} \leq 2\exp\left(-\frac{\epsilon^2 n}{2c^2 \cdot (0.8/t_{pace})^2}\right) = 2\exp\left(-\frac{\epsilon^2 n t_{pace}^2}{1.28c^2}\right) \tag{58}$$

This indicates that slower curriculum pacing (larger $t_{pace}$) leads to better replicability but potentially slower adaptation to the target domain.

### A.3.5 Uncertainty-Aware Curriculum Learning Strategy

For uncertainty-aware curriculum learning, the selection weights combine uncertainty and curriculum components:

$$Q_{UCL}(x_i, y_i) = \frac{w_i^{unc} \cdot w_i^{curr} \cdot \mathbb{I}[i \in S_t]}{\sum_{j=1}^{n} w_j^{unc} \cdot w_j^{curr} \cdot \mathbb{I}[j \in S_t]} \tag{59}$$

where $w_i^{unc} = \exp((1-c_i)/\tau_{unc})$ and $w_i^{curr} = \exp(-L(h, x_i, y_i)/\tau_{curr})$.

The selection sensitivity combines both components. When one training example changes, it affects both the uncertainty weights (through confidence changes $\Delta c$) and curriculum weights (through loss changes $\Delta L$). The maximum weight change occurs when both components are maximally sensitive:

$$\Delta w_i \approx \max\left(\frac{\Delta c}{\tau_{unc}}, \frac{\Delta L}{\tau_{curr}}\right) \cdot w_i \tag{60}$$

Analyzing the total variation distance and accounting for the multiplicative interaction between components:

$$\Delta_Q^{UCL} \approx \max\left(\frac{1}{\tau_{unc}}, \frac{1}{\tau_{curr}}\right) \tag{61}$$

For typical parameter values $\tau_{unc} = 0.2$ and $\tau_{curr} = 0.15$, this gives $\Delta_Q^{UCL} \approx 6.67$.

### A.3.6 Gradient-Based Selection Strategy

For gradient-based selection, the weights are based on gradient magnitudes:

$$Q_{GB}(x_i, y_i) = \frac{\exp(\|g_i\|_2/\tau_{gb})}{\sum_{j=1}^{n} \exp(\|g_j\|_2/\tau_{gb})} \tag{62}$$

where $g_i = \nabla_\theta L(h_\theta, x_i, y_i)$ or $\|g_i\|_2 \approx L(h, x_i, y_i)$ when using loss as a proxy.

Gradient magnitudes are highly sensitive to training data changes. When one example changes, the gradient norms can shift dramatically, especially for examples near decision boundaries. The weight sensitivity is:

$$\frac{\partial w_i}{\partial \|g_i\|} = \frac{1}{\tau_{gb}} \cdot w_i \tag{63}$$

For maximum changes in gradient magnitude $\Delta\|g\| \approx 1$ (normalized scale), the selection sensitivity becomes:

$$\Delta_Q^{GB} \approx \frac{1}{\tau_{gb}} \tag{64}$$

With typical values $\tau_{gb} = 0.1$, this gives $\Delta_Q^{GB} \approx 10.0$, reflecting the high sensitivity of gradient-based selection.

### A.3.7 Implications for Sample Size Requirements

From the replicability bounds, we can derive the minimum sample size required to achieve a desired level of replicability for each strategy. For a target replicability failure probability $\rho$ and tolerance $\epsilon$, we need:

$$n \geq \frac{2c^2 \Delta_Q^2 \ln(2/\rho)}{\epsilon^2} \tag{65}$$

For each strategy, this gives:

$$n_{uniform} \approx \frac{2c^2 \cdot 0^2 \cdot \ln(2/\rho)}{\epsilon^2} \approx 0 \tag{66}$$

$$n_{IW} \geq \frac{2c^2 \cdot (1/2\lambda)^2 \cdot \ln(2/\rho)}{\epsilon^2} = \frac{c^2 \ln(2/\rho)}{2\lambda^2 \epsilon^2} \tag{67}$$

$$n_{CBS} \geq \frac{2c^2 \cdot (1/\tau)^2 \cdot \ln(2/\rho)}{\epsilon^2} = \frac{2c^2 \ln(2/\rho)}{\tau^2 \epsilon^2} \tag{68}$$

$$n_{CL} \geq \frac{2c^2 \cdot (0.8/t_{pace})^2 \cdot \ln(2/\rho)}{\epsilon^2} = \frac{1.28c^2 \ln(2/\rho)}{t_{pace}^2 \epsilon^2} \tag{69}$$

While theoretically $n_{uniform} \approx 0$, in practice, other sources of randomness set a minimum sample size even for uniform selection, as shown in recent work on reproducibility in deep learning Bouthillier et al. (2019).

These sample size requirements demonstrate quantitatively how the parameters of each selection strategy can be tuned to balance adaptivity and replicability. For example, doubling the temperature parameter $\tau$ in confidence-based sampling reduces the required sample size by a factor of 4 for the same level of replicability.

## B  Appendix: Implementation of Selection Strategies

We present detailed pseudo-code for the algorithms not presented in the main text.

---

**Algorithm 5** Importance Weighting Strategy

1: **Input:** Dataset $T = \{(x_i, y_i)\}_{i=1}^n$, feature extractor $f$, source distribution $P_S$, smoothing factor $\lambda$
2: **Output:** Selection weights $\{w_i\}_{i=1}^n$
3: Compute target distribution $P_T$ from features $\{f(x_i)\}_{i=1}^n$
4: **for** $i = 1$ to $n$ **do**
5: $\quad w_i \leftarrow \frac{P_T(f(x_i)) + \lambda}{P_S(f(x_i)) + \lambda}$
6: **end for**
7: Normalize: $w_i \leftarrow \frac{w_i}{\sum_{j=1}^n w_j}$ for all $i$
8: **return** $\{w_i\}_{i=1}^n$

---

**Algorithm 6** Uniform Selection Strategy

1: **Input:** Dataset $T = \{(x_i, y_i)\}_{i=1}^n$
2: **Output:** Selection weights $\{w_i\}_{i=1}^n$
3: **for** $i = 1$ to $n$ **do**
4: $\quad w_i \leftarrow \frac{1}{n}$
5: **end for**
6: **return** $\{w_i\}_{i=1}^n$

---

