# OpenReview forum: "Sensitivity of Stability: Theoretical & Empirical Analysis of Replicability for Adaptive Data Selection in Transfer Learning"
_TMLR — Rejected by TMLR_

### Review · Reviewer_G76q · 2025-11-04

**Summary Of Contributions:**

This paper presents a theoretical and empirical analysis of replicability in transfer learning, focusing on how adaptive data selection strategies impact the consistency of results.


To analyze this problem, the author introduce a formal measure called selection sensitivity (∆Q), which quantifies how much a data selection strategy changes in response to small perturbations in the training data. Their theoretical analysis reveals that the probability of replicability failure increases quadratically with this selection sensitivity but decreases exponentially as the sample size grows.

Through experiments on the MultiNLI dataset, the study confirms this relationship. Strategies with high sensitivity, like gradient-based selection, achieve the highest accuracy but also suffer from very high replicability failure rates. In contrast, less adaptive strategies like uniform sampling are more consistent, with failure rates as low as 2.22%, but yield lower accuracy. Moreover, the paper shows that pre-training on the source domain before fine-tuning on the target domain serves as a powerful mitigation technique, significantly improving both performance and replicability for all strategies.

**Audience:**

Yes

**Audience Explanation:**

This paper concludes a trade-off between performance and replicability of data selection strategy in transfer learning. While this result could be intuitive, the bridge introduced by this paper, i.e., sensitivity, is interesting and could be of value.

**Claims And Evidence:**

No

**Claims Explanation:**

There are mainly two concerns:

First, the authors claim that the theoretical bound is tight, but the reviewer could not agree. While it might be tight from Lemma 1 to Theorem 1, it is very easy to tell Lemma 1 itself is a very loose bound.

While the first concern might be argued as subjective, the second concern is vital. In figure 2(b), how could the four points, i.e., uniform, importance weighting, confidence sampling, curriculum learning, locate above the theoretical bound. As the red line is already the upper bound of $\rho$, there must be technical errors in at least one of the components in this paper: proof of the bound, calculation of sensitivity, experiment implementation.

**Requested Changes:**

Please refer to above concerns.

---

### Review · Reviewer_5bdZ · 2025-11-15

**Summary Of Contributions:**

This paper investigates the replicability of transfer learning methods based on adaptive data selection. Specifically, the authors introduce a new notion called *selection sensitivity*, which characterizes the behavior of data selection methods—that is, weighting schemes applied to training data. They show that this selection sensitivity effectively explains an upper bound on replicability. The main results are supported not only by theoretical analysis but also by empirical validation through numerical experiments.

**Audience:**

Yes

**Audience Explanation:**

Replicability and transfer learning are both major topics of interest within the machine learning community, and I believe the content of this paper has the potential to attract considerable attention from many researchers.

**Claims And Evidence:**

No

**Claims Explanation:**

I have concerns about the validity of **Lemma 1**, which plays a crucial role in proving the main result, Theorem 1. In particular, it is unclear to me how inequality **(27)** is derived. It appears to be claimed that (27) follows from (26). Namely, the argument seems to rely on the idea that if the gap between the gradient functions used by the two algorithms is uniformly bounded by $\epsilon$, then the difference between the outputs of the corresponding gradient descent procedures can be bounded by $\eta E \epsilon$, where $E$ is the number of steps and $\eta$ is the step size. However, once we take into account the accumulation of errors in gradient descent, I do not believe such a bound holds in general.

For example, consider the following counterexample:

* $\Theta = \mathbb{R}$, i.e., the parameter is one-dimensional;
* the initial point is $\theta^{(1)} = 0$;
* one gradient function is $\nabla R(\theta) = -\theta$, and the other is $\nabla R'(\theta) = -\theta - \epsilon$.

Then, for any $\theta$, we have
$$
|\nabla R(\theta) - \nabla R'(\theta)| \le \epsilon,
$$
so the condition corresponding to (26) is satisfied. In this case, gradient descent based on $\nabla R$,
$$
\theta^{(t+1)} = \theta^{(t)} - \eta \nabla R(\theta^{(t)}),
$$
with the initial value $\theta^{(1)} = 0$, yields $\theta^{(t)} = 0$ for all $t$. On the other hand, for gradient descent based on $\nabla R'$, we have $\theta^{(2)} = \eta \epsilon$ and
$$
\theta^{(t+1)} = \theta^{(t)} + \eta(\theta^{(t)} + \epsilon) \ge (1+\eta) \theta^{(t)},
$$
which implies
$$
\theta^{(t)} \ge \eta \epsilon (1 + \eta)^{t - 2}.
$$
Thus, the difference between the final outputs can be at least
$$
\eta \epsilon (1 + \eta)^{E - 2},
$$
which can be much larger than the upper bound $\eta E \epsilon$ claimed in (27).

---

I also have concerns about the validity of the reported values of the **selection sensitivity** in Table 1 and Table 2 in the experimental section. Although the computation method is described in the appendix, it is based on very rough approximations rather than exact calculations. For example, right after equation (63), the authors introduce a strong assumption that the gradient norm is roughly equal to one.
It is not clear whether this assumption is justified under the actual experimental setups used in the paper.

In particular, several important constants in the selection sensitivity calculation appear to be chosen somewhat ad hoc as “typical values.” For instance, the typical value of $\lambda$ in (47) is set to 0.1, and similar substitutions are made in (56), (61), and (64). From my reading, I could not determine whether these values actually correspond to the configurations used in the numerical experiments. A more detailed explanation of the rationale behind these choices—especially their relevance to the experimental settings—would be helpful.

Finally, I recommend clearly stating in the main text that the selection sensitivity values reported in the tables are only rough estimates. Since they appear in the “Numerical Experiments” section, readers might otherwise mistakenly interpret them as quantities obtained via precise numerical computation.

**Requested Changes:**

I would appreciate it if the authors could review the concerns I raised above regarding the evidence, and either provide responses and revisions accordingly or point out any misunderstandings on my part.

---

### Review · Reviewer_H8z6 · 2025-11-30

**Summary Of Contributions:**

The paper „Sensitivity of Stability: Theoretical & Empirical Analysis of Replicability for Adaptive Data Selection in Transfer Learning“ explores how adaptive data selection strategies influence the training of deep neural networks, especially their replicability. The main question is: what are the differences in performance when training the same neural network multiple times with different seeds: will the adaptive data selection strategies result in substantially different outcomes? For this, the paper derives an upper bound on the difference between two neural network training runs if the dataset only differs by one training sample, and then uses this result to bound the probability of arriving at the same accuracy when running the neural network transfer learning multiple times. The theoretical setup is validated via an experiment, where six different strategies for selecting samples rank in terms of replicability as predicted by theory. The original setup, that could also be phrased finetuning, is complemented by a second setup where there is a continued pre-training that leads to better results, but for which no theory exists.

## Strengths:
* Novel theory on adaptive data selection for transfer learning (although the setting could also be phrased as finetuning).
* Experimental result showing how finetuning can be improved by continued pretraining on a related source task
## Weaknesses:
* Structure and clarity of the paper need to be improved substantially. Currently, several key metrics are undefined, and the exact training procedure is not clear either.
* experiments should be extended to a setting with varying dataset sizes.

**Additional Comments:**

* Are the Qs in Equation 18 and 19 related to each other?
* The paper hypothesizes that the higher batch size leads to sharper local minima. This is a hypothesis that could be tested, maybe in future work? It would be interesting to see if the continued retraining with the source task has an influence on the sharpness and generalisation?
* From a visual inspection of Figure 3 I conclude that curriculum is more stable than uniform: it has a lower spread of final values and the lines are closer, which should result in a lower std for curriculum. However, the results in Table 1 disagree. It would be great if the authors could clarify this.
* Do we train using the three categories at a time or one? And does the evaluation set also contain the same three categories or only one? The paper needs to be more thorough here.
* Algorithm 5: are the weights per category constant?

**Audience:**

Yes

**Audience Explanation:**

Even though the paper appears to be on transfer learning, the described setup rather resembles finetuning of a foundation model: a pretrained model is equipped with a new head and the weights are updated for a small dataset. This setup is extremely common and any improvement in this setup should be of interest to most readers of TMLR.

**Claims And Evidence:**

Yes

**Claims Explanation:**

The paper delivers on most claims. The only claims I think is not fulfilled is code release: there is no code for reviewing, which limits my trust in the results and this claim.

**Requested Changes:**

## Required:
* Properly introduce the metrics used in Tables 1 and 2 (empirical replicability failure rate). I think they are introduced in the section afterwards (Section 5.2), which makes reading Section 5.1 very confusing.
* Explain sources in uncertainty in the experiments, I.e., what is the influence of the random seeds? Do they only affect the sampling of data points or also weight initialisation and data shuffling for SGD?
* Sensitivity analysis for epsilon. How does the size of epsilon impact the replication failure?
* Section 5.1, Page 13, 2nd Paragraph describes that the observations are closed to the calculated curve. However, I cannot find such curves. I assume this would be similar to a scaling law and should be displayed. This is related to the 3. enumeration item in the discussion, which states that the paper gives practical guidance on the amount of required data for desired replicability levels. However, I think this is overstating the contribution, this is merely a rough guide, and I would appreciate a more thorough study (plot) that shows the stability as a function of the data set size. This can also be motivated by the fact that typical finetuning setups have less data available (this paper uses 6000 samples).
* Figure 4 needs to be properly explained. There should be an explanation how the min and max are displayed, and what can be seen in the figure on the left-hand side.
* Code
## Minor:
* Training pseudo-code would be great. I don‘t know if the data is only re-weighted according to the strategies, or sampled differently for the mini-batches. This is an important difference as exemplified by the ongoing research on random reshuffling versus sampling with replacement (see for example “Random Reshuffling is Not Always Better” by Christopher M De Sa.
* The paper should be more careful in the use of cite, a lot of references just randomly appear in the middle of a sentence, breaking the reading flow.
* Stating the number of random seeds in Section 5 is a bit repetitive
*I am concerned about this paper mostly standing on the shoulders of arXiv papers. Are these really all arXiv papers, or are some of them already published? TBH both cases are concerning, in one case I am wondering what it means that the submission has to draw so heavily on unpublished work. In the other case, I would be concerned by the sloppiness of the authors in preparing the references and wonder if the authors were also sloppy in other parts of the submission.
* Bouthilier et al. (2019) is only a secondary source for replicability. Please also cite the original sources mentioned in Bouthilier et al. (2019).

---

### Decision · Action_Editor_MdHU · 2026-01-05

**Recommendation:** Reject

**Additional Comments:**

Reviewers unanimously recommend to reject. Given the authors did not send a rebuttal it's a straightforward reject.

**Audience:**

Yes

**Audience Explanation:**

The topic of the submission is relevant to the TMLR audience.

**Claims And Evidence:**

No

**Claims Explanation:**

The reviewers pointed out concerns about technical contents of the submission and the authors did not respond.